# Impact of *CYP2C9* and *VKORC1* Polymorphisms on Warfarin Sensitivity and Responsiveness in Jordanian Cardiovascular Patients during the Initiation Therapy

**DOI:** 10.3390/genes9120578

**Published:** 2018-11-27

**Authors:** Laith N. AL-Eitan, Ayah Y. Almasri, Rame H. Khasawneh

**Affiliations:** 1Department of Applied Biological Sciences, Jordan University of Science and Technology, Irbid 22110, Jordan; ayalmasri13@sci.just.edu.jo; 2Department of Biotechnology and Genetic Engineering, Jordan University of Science and Technology, Irbid 22110, Jordan; 3Department of Hematopathology, King Hussein Medical Center (KHMC), Jordan Royal Medical Services (RMS), Amman 11118, Jordan; rami.khasawneh@jaf.mil.jo

**Keywords:** *CYP2C9*, *VKORC1*, warfarin, warfarin initiation phase of therapy, INR, pharmacogenetics study

## Abstract

Warfarin is an oral anticoagulant frequently used in the treatment of different cardiovascular diseases. Genetic polymorphisms in the *CYP2C9* and *VKORC1* genes have produced variants with altered catalytic properties. A total of 212 cardiovascular patients were genotyped for 17 Single Nucleotide Polymorphisms (SNPs) within the *CYP2C9* and *VKORC1* genes. This study confirmed a genetic association of the *CYP2C9**3 and *VKORC1* rs10871454, rs8050894, rs9934438, and rs17708472 SNPs with warfarin sensitivity. This study also found an association between *CYP2C9* and *VKORC1* genetic haplotype blocks and warfarin sensitivity. The initial warfarin dose was significantly related to the *CYP2C9**3 polymorphism and the four *VKORC1* SNPs (*p* < 0.001). There were significant associations between rs4086116 SNP and TAT haplotype within *CYP2C9* gene and rs17708472 SNP and CCGG haplotype within *VKORC1* gene and warfarin responsiveness. However, possessing a *VKORC1* variant allele was found to affect the international normalized ratio (INR) outcomes during initiation of warfarin therapy. In contrast, there was a loose association between the *CYP2C9* variant and INR measurements. These findings can enhance the current understanding of the great variability in response to warfarin treatment in Arabs.

## 1. Introduction

Warfarin is a commonly prescribed oral anticoagulant that is employed for the treatment of venous and arterial thromboembolic disorders and cardiac valve replacements [1]. However, interindividual genetic variation causes great variability in dosage requirements, making the latter a problematic issue for physicians. A higher or lower dose than needed could lead to bleeding and thrombotic risk, respectively [2,3]. Two-thirds of warfarin dose variation was due to environmental factors like age, body mass index, smoking status, gender, and diet, among others, while the remaining one-third is caused by genetic factors such as the *CYP2C9* and *VKORC1* genes [4,5,6].

Belonging to the cytochrome P450 superfamily, the *CYP2C9* gene is involved in the metabolism and clearance of S-warfarin, the latter of which is a racemic form of warfarin together with R-warfarin [7,8]. *CYP2C9* is located on the long arm of chromosome 10, and like other members of *CYP2C*, *CYP2C9* is highly polymorphic [9,10,11]. Although there are over 50 single nucleotide polymorphisms (SNPs) located in the regulatory and coding region of the *CYP2C9* sequence, the most studied *CYP2C9* polymorphisms are *CYP2C9*2* (R144C) and *CYP2C9*3* (I359L) [12,13]. It was found that individuals carrying the *CYP2C9* *2 *3 alleles reduced the elimination of S-warfarin and, therefore, plasma concentrations of the latter increased significantly compared to the wild-type allele and the individuals with these variant alleles need a lower warfarin maintenance dose [14].

Similarly, the *VKORC1* gene encodes the vitamin K epoxide reductase complex subunit 1 and it is a warfarin target [5]. The vitamin K epoxide reductase complex subunit 1 normally catalyzes the carboxylation reaction of the vitamin K-dependent protein glutamic acid residues in order to activate it, the latter of which are responsible for catalyzing the clotting factor pathway [15]. Several genetic studies conducted in different populations suggest that the G3673A (rs9923231), C6484T (rs9934438), and G9041A (rs7294) polymorphisms of the *VKORC1* gene are the most common and well-studied [7,16].

Analysis of *CYP2C9* and *VKORC1* gene polymorphisms revealed that they were responsible for 10% and 40% of warfarin dose requirement variance, respectively [17]. Combined with clinical data, both of the aforementioned genes can explain up to 60% of the warfarin variance [18,19]. More than 50 years ago, the dose of warfarin was determined by trial and error, with an initial dose (2–10 mg/day) dependent on the indication of warfarin and clinical factors, regardless of the effect of the genetic factor [20]. Patients are treated with warfarin in two stages, the first called the initiation phase of treatment, which is considered as the stage in which the initial international normalized ratio (INR) value of the patient is unstable and fluctuates up and down. With the second (maintenance) stage of the therapy, the patient is within the therapeutic INR for at least two consecutive visits [21]. However, pharmacogenetics for specific populations including Jordanian Arabs is necessary. Therefore, the objective of this study was to recognize genetic variations within the *CYP2C9* and *VKORC1* genes that are involved in warfarin sensitivity and responsiveness in Jordanian cardiovascular patients of Arab descent during initiation of treatment.

## 2. Materials and Methods

### 2.1. Patient Population and Study Design

The study population consisted of 212 unrelated warfarin intake patients selected from the Jordanian-Arab population, from the Anticoagulation Clinic at the Queen Alia Heart Institute (QAHI) in Amman, Jordan. Informed consent was obtained from all subjects. The study protocol was approved by the Human Ethics Committee at Jordan University of Science and Technology in Irbid, Jordan, and the Royal Medical Services in Amman, Jordan. Ethical approval code: 13/78/2014.

In this study, patients have cardiovascular diseases and are prescribed warfarin as an anticoagulant therapy. Inclusion criteria involved patients being 18 years or older and having received warfarin for at least three months. Patients who did not provide informed written consent, did not visit the anticoagulation clinic regularly, took *CYP2C9* inducers or inhibitors, or did not have a complete data set were excluded. 

Initially, 350 patients were screened and, based on the aforementioned inclusion and exclusion criteria, 300 patients were approached to participate in this study (Figure 1). 80 patients were subsequently excluded because of inability to complete the treatment program or refusal. Of the remaining patients, 220 accepted to be part of this study, after which an additional eight patients were excluded from the final analysis because of a failure in genotyping. In total, whole sets of data were obtained from 212 patients with cardiovascular disease who were being treated with warfarin. Data was collected on demographic (age, gender, and body mass index) and lifestyle (smoking status and diet) characteristics as well as medical history (diseases and clinical features of warfarin therapy). Clinical features included the target INR, mean weekly warfarin doses required to reach the target INR, and the use of concomitant medication, all of which are required to be known in order to adjust warfarin doses. All data were blinded and obtained through semi-standardized interviews and medical records. 

All subjects in this study received warfarin anticoagulant therapy according to RMS anticoagulation protocol, which began with 2.5 to 10 mg nightly doses. INR monitoring is required at least once a week for the first three to four weeks after the initiation of therapy. After three consecutive visits, a patient with stable INR reaches the maintenance dose and is monitored for warfarin treatment administration by clinic protocol.

### 2.2. Outcome Measurement 

Oral anticoagulant therapy was mandated by the prothrombin time (PT) that is, evaluated using an automated method over STAGO coagulometric unit in the QAHI laboratory. To calculate INR, there was a blood coagulation (clotting) test. INR monitoring for dose adjustments was determined by the physician and pharmacist. INR values between January 2014 and November 2015 were obtained from medical records, and these values were then used to divide the patients according to warfarin responsiveness into: (1) good responders, with an INR value within the target range; (2) poor responders, with an INR value under the target range; and (3) extensive responders, with an INR above the target range.

Further, based on their warfarin sensitivity, patients were divided into resistant, normal, or sensitive to warfarin as follows: (1) warfarin resistance (or Poor metabolizer), largest daily doses were required to keep a patient’s INR within a therapeutic range (dose required > 49 mg/week); (2) warfarin normal patient (or Intermediate metabolizer), intermediate doses required (doses between 21–49 mg/week); and (3) warfarin sensitive patient (or Extensive metabolizer) lowest doses required (required dose <21 mg/week) [22].

### 2.3. SNP Selection, DNA Extraction, and Genotyping

In this study, 17 SNPs in the *CYP2C9* and *VKORC1* genes were selected from public databases and genotypes. Information about the aforementioned SNPs is shown in Appendix A. Genomic DNA was extracted within one week of blood collection using the commercially available Wizard Genomic DNA Purification Kit (Promega Corporation, Madison, WI, USA) according to the manufacturer’s instructions. After extraction, the DNA was diluted in 96-well plates using an automated robotic system to achieve concentrations of 20 ng/μL (50–500 μL). Concentrations were confirmed with the Nano-Drop ND-100 (Thermo Scientific, Wilmington, DE, USA). Genotyping was carried out by means of the MassARRAY® system (iPLEX GOLD) (Sequenom, San Diego, CA, USA), which was carried out at the Australian Genome Research Facility (AGRF) (Sequenom).

### 2.4. Statistical Analysis 

Discrepancy and call rates were calculated using Microsoft Excel, and the deviation from Hardy Weinberg Equilibrium (HWE) was assessed using the Pearson X2 test. Minor allele frequencies (MAF) and HWE *p*-values for genotypic distribution were calculated via the Court lab-HW calculator. To test which of the chosen SNPs is associated with warfarin response, various statistical genetic association analyses were conducted, such as the chi-square, nonparametric correlation tests (Kruskal-Wallis and Tukey Pairwise comparison) and haplotype genetic analysis test. The Statistical Package for the Social Sciences (SPSS) version 21.0 and the SNPStat Web Tool (https://www.snpstats.net/start.htm) were used to perform all analyses.

## 3. Results

### 3.1. Study Group

The study group comprised of 212 unrelated Jordanian-Arabs patients treated with warfarin, with a mean age (±SD) of 56.03 (±17.68) years, a median age of 60, and an age range of 18 to 85 years. There were 34 poor responders (16%), 146 moderate responders (68.9%), and 32 extensive responders (15.1%). Table 1 summarizes the demographic, lifestyle, and medical characteristics of each of the three groups.

In total, 17 SNPs (100%) passed the quality control measures for throughput genotyping and were analyzed by the MassARRAY® system (iPLEX GOLD) with high accuracy and a 97% average success rate. The genotypic discrepancy average (±SD) rate over the 17 loci was only 0.06% (±0.0004%) out of the entire cohort (212 subjects). Genotypic and allelic frequencies are shown in Appendix A.

For the 17 SNPs examined in this study, all were in accordance with the HWE. Ten polymorphisms (rs104894539, rs104894540, rs104894541, rs104894542, and rs61742245 in *VKORC1*, and rs28371685, rs28371686, rs72558191, rs9332131, and rs9332239 in *CYP2C9*) were non-polymorphic. In contrast, the seven remaining SNPs (rs10871454, rs8050894, rs9934438, and rs17708472 located in *VKORC1*, and rs1799853, rs4086116, and rs1057910 located in *CYP2C9*) were polymorphic and thus included in the study. The minor alleles and their frequencies for the successful genotyped SNPs are shown in Appendix A.

### 3.2. Effect of *CYP2C9* and *VKORC1* Polymorphisms on Warfarin Sensitivity during Initiation Phase of Therapy

Regarding the association of *VKORC1* and *CYP2C9* SNPs with warfarin sensitivity among the three inclusion groups, significant differences in proportions among genotypes were observed at all tested *VKORC1* SNPs (*p* < 0.001) (Table 2). Significant differences were also observed between two SNPs of the *CYP2C9* gene (rs4086116 (*p* = 0.012) and rs1057910 (*p* < 0.001)), as shown in Table 2. Moreover, there was a significant association observed between *VKORC1* and *CYP2C9* haplotypes and warfarin sensitivity (*p* < 0.0001) (Table 3).

### 3.3. Effect of *CYP2C9* and *VKORC1* Polymorphisms on Warfarin Required Dose during Initiation Phase of Therapy

Carriers of *CYP2C9* and *VKORC1* polymorphisms had a significantly increased required dose compared with wild-type subjects or carriers of only one polymorphism of *CYP2C9* or *VKORC1* (Table 4).

### 3.4. Effect of *CYP2C9* and *VKORC1* Polymorphisms on Warfarin Responsiveness during Initiation of Therapy

There were no significant differences in patient responder groups regarding the *VKORC1* and *CYP2C9* SNPs except for the *VKORC1* rs17708472 (*p* = 0.042) and the *CYP2C9* rs4086116 (*p* = 0.005) SNPs (Table 5). However, significant associations were found between genetic haplotypes of CCGG *VKORC1* and TAT *CYP2C9* and warfarin sensitivity, with *p* = 0.02 and *p* = 0.018, respectively (Table 6).

### 3.5. Effect of *CYP2C9* and *VKORC1* Polymorphisms on INR Treatment Outcome 

There were no significant differences observed between the *CYP2C9* SNP genotypes and INR values measured at start of treatment for 212 cardiovascular patients treated with warfarin. In contrast, significant differences were observed between INR values measured at the initiation phase of therapy and certain *VKORC1* SNPs, namely rs10871454 (*p* = 0.006), rs8050894 (*p* = 0.007), and rs9934438 (*p* = 0.009), as shown in Table 4.

### 3.6. Correlation Between Warfarin Dose and Clinical Data

Finally, there was no significant correlation between warfarin dose and body mass index, age, gender, co-morbidities, or the treatment indication (*p* = 0.505).

## 4. Discussion

Earlier studies on warfarin pharmacogenetics provide evidence that common *VKORC1* and *CYP2C9* polymorphisms with clinical and environmental factors are responsible for over half of the variability in warfarin required dose [23,24]. Genotyping patients who are carriers of *VKORC1* and *CYP2C9* variant alleles has been proven to reduce the risk of over-anticoagulation compared to the traditional initial dose approach [25,26].

In this study, our goal was to identify genetic factors associated with sensitivity and responsiveness to warfarin treatment during the initiation phase of treatment in Jordanian-Arab patients with cardiovascular disease. The results of the current pharmacogenetic study strongly suggest that there is a significant association of the *VKORC1* rs8050894, rs10871454, rs9934438, and rs17708472 SNPs and the *CYP2C9* rs4086116 and rs1057910 SNPs and their haplotypes with the required warfarin dosage and warfarin sensitivity. This study also reported that there is a genetic association between *VKORC1* rs17708472 SNP and CCGG genetic haplotype block and *CYP2C9* rs4086116 SNP and TAT genetic haplotype block with warfarin responsiveness during the initiation phase of therapy.

The allelic frequencies of *CYP2C9* and *VKORC1* SNPs in our population were similar to those found in other ethnic groups, as in the case of *CYP2C9*2*, with 10% in our population, 6% in American and European populations, and 1% in Africans [27]. However, allelic frequencies of *VKORC1* SNPs were found to differ drastically from other populations. For example, rs10871454 was 52% in our population compared to 41% in Americans, 39% in Europeans, and 6% in African populations [27]. With regard to the association of *CYP2C9* polymorphisms with warfarin sensitivity, our results are consistent with the study by Takahashi et al. (2001) which shows that *CYP2C9* *2 and *3 polymorphisms reduce warfarin clearance [28] as *CYP2C9* is the major metabolizing enzyme of warfarin, therefore, reduction of activity results in lower required doses needed to achieve the therapeutic INR. We found a strong association of *CYP2C9*3* (rs1057910 A>C) and *CYP2C9* (rs4086116 C>T) genotypes with warfarin sensitivity during the initiation stage of treatment with *p* < 0.001 and *p* = 0.012, respectively (Table 2). This study reported that individuals with one variant allele were associated with an increased risk of warfarin sensitivity. For example, 41.5% of the patients who carried the rs1057910 A>C variant allele were sensitive to warfarin, compared to 8.8% of the wild-type patients were sensitive. Moreover, carrying a *CYP2C9* TCC genetic haplotype block was significantly associated with warfarin sensitivity with *p* < 0.0001 (Table 3).

In disagreement with other studies, the *CYP2C9*2* variant did not show a significant association with warfarin sensitivity (*p* = 0.744). This can possibly be explained by the genotypic frequency having an impact on the association; in our population only two patients were homozygous for the variant allele (TT), 32 were heterozygous patients (CT), and 105 patients were homozygous for the wild-type allele (CC) (Table 2). Moreover, it has been proposed that patients who carry one *CYP2C9* 2* allele results in a dose reduction compared with the wild-type dose [14,29,30]. Although the allelic frequency of *CYP2C9*2* (rs1799853) in our samples was in agreement with the American and European populations [27], this study did not find significant differences between this SNP and variability in required doses in the initiation phase of therapy (*p* = 0.366) as shown in Table 4 and Figure 2.

Clinical pharmacogenetic studies suggested that patients who carry the *CYP2C9*3* (rs1057910) C allele leads to a dose reduction of 28–41% [12,29,30] in the Caucasian American population and from 12–38% of the Asian population compared to the wild-type [31,32,33,34]. In alignment with the aforementioned studies, we revealed that there is a 34.3% reduction in warfarin dose. Patients carrying one variant allele (C) required 41.75 mg/week, in comparison with the wild-type allele which required 27.43 mg/week (*p* < 0.001), as shown in Table 4 and Figure 2. Therefore, in our study the *CYP2C9*3* allele has a greater effect on variation in warfarin dose during the initiation phase of therapy compared with *CYP2C9*2*. In the case of individuals carrying rs4086116 C>T variant allele, this resulted in 23.9% and 37.9% reduction on warfarin dose compared to wild-type with *p* = 0.016, as shown in Table 4 and Figure 2.

For the four studied *VKORC1* SNPs, we observed a strong association of *VKORC1* SNPs with warfarin sensitivity (*p* < 0.001). For example, patients with the T allele for rs10871454 C>T showed a high risk of warfarin sensitivity with 25.5% (CT) and 48.3% (TT) reduction of the required dose, respectively. In this case, the drug target enzyme could be expressed in smaller amounts and, therefore, low doses of the drug can obtain a therapeutic INR in an initial phase of therapy (Table 2). Furthermore, *VKORC1* genetic haplotype analysis showed a significant association between three *VKORC1* genetic haplotype blocks and sensitivity to warfarin with *p* < 0.0001 (Table 3). Moreover, Schelleman et al. (2007) and Wadelius et al. (2005) reported that a correlation exists between *VKORC1* SNP 1173 C> T (rs9934438), and the variation in warfarin dose patients carrying the variant allele of this SNP is related to a reduction in the required dose compared to the wild-type [5,35]. In alignment with these results, we found that patients who carry one variant allele required an average dose of 39.47mg/week and two variant allele carriers needed an average of 26.82 mg/week, while patients carrying the wild type CC needed an average dose of 53.02 mg/week with *p* < 0.001 as shown in Figure 3. Limdi et al. (2007) and Shrif et al. (2011) showed that *VKORC1* rs8050894 (1542G>C) were associated with lower warfarin doses in European Americans and Sudanese patients, respectively [36,37]. Accordingly, our results found a significant association between lower required warfarin dose and this SNP at an initial phase of therapy with *p* < 0.001 as shown in Figure 3.

For the last *VKORC1* SNP *VKORC1**4 C<T (rs17708472), our study is similar to the study by Haug et al. (2008), which reported that this SNP was associated with higher dose requirements [38]. Our results showed that this variant was associated with significant differences in initial warfarin required dose; patients who were homozygous for the variant allele (TT) genotype required an average dose of 57.8 mg/week, heterozygous (CT) patients required an average dose of 41.63 mg/week, while wild-type (CC) patients required an average dose of 33.65 with *p* < 0.001 (Table 4).

With regard to the correlation of *CYP2C9* and *VKORC1* SNPs and warfarin responsiveness, we compared SNP genotypes with the warfarin responder groups (poor, good, and extensive responders). Significant differences were found between *VKORC1* rs17708472 (C>T) genotypes and the three different responder groups (Table 5); 33.3% of the patients carrying the variant allele (TT) were within the extensive responder group (meaning this variant allele was associated with increased risk of over-anticoagulation), compared to 11.3% of the wild-type (CC) patients who were extensive responders (*p* = 0.042). Accordingly, Kringen et al. (2011) have also shown that patients who carry this SNP are associated with an increased risk of the existence of therapeutic INR (over-anticoagulation) [39]. Otherwise, *VKORC1* rs10871454, rs8050894, and rs9934438 alleles show no significant differences between SNP genotypes within the three responder groups in our population, with *p* = 0.171, 0.235, and 0.226, respectively (Table 5). In contrast, *VKORC1* CCGG genetic haplotype block showed a significant association with warfarin responsiveness with *p* = 0.02 (Table 6).

Moreover, significant differences were observed between *VKORC1* rs10871454, rs8050894, and rs9934438 SNPs and INR value during the initiation phase of therapy (*p* = 0.006, 0.007, and 0.009, respectively), while rs17708472 SNP showed no significant differences (*p* = 0.493). Therefore, in our population the *VKORC1* SNP genotypes are associated with the generation of a high or low INR during the initiation phase of therapy (Table 4).

Taube et al. (2000) reported that an individual carrying an allelic variant of *CYP2C9* was not associated with an increased incidence of severe over-coagulation during long-term treatment [40]. Correspondingly, our results show no significant differences between the *CYP2C9*2* and *3 genotypes within the three responder groups with *p* = 0.076 and 0.910, respectively (Table 5). Conversely, our study reported significant differences between the proportion of *CYP2C9* rs4086116 (C>T) genotype and the three different responder groups; 50% of TT carriers were within the extensive responder group compared with 10.6% of wild type CC carriers (*p* = 0.005), which means that TT carriers are associated with increased risk of over-anticoagulation (Table 5). Moreover, significant association was observed between *CYP2C9* TAT genetic haplotype block and warfarin responsiveness with *p* = 0.018 (Table 6).

In addition, we did not observe significant differences in the three studied SNPs (*CYP2C9*2, *3,* and *CYP2C9* (C> T) rs4086116) and the INR value during the initiation phase of therapy. Therefore, in our population *CYP2C9* is not associated with the generation of a high or low INR as shown in Table 4.

Confirmation of our results and ongoing research including additional factors will be accomplished in a larger patient cohort, including genetic factors such as *OATP* transporters (mediates the uptake of warfarin into hepatocytes), *CYP3A4*, *CYP1A1*, and *CYP1A2* enzymes (metabolizing of R-warfarin), or *GGCX* encoded *gamma-glutamyl carboxylase* (the reduced vitamin K–form to activate coagulation factors) [5,41]. Application for individualized warfarin treatment will be both beneficial and efficient for cardiovascular patients in the future. Finally, the majority of the population included in this study is elderly, with only 15% of the subjects under 40 years of age. Therefore, additional study is needed in children and young adults.

## Figures and Tables

**Figure 1 genes-09-00578-f001:**
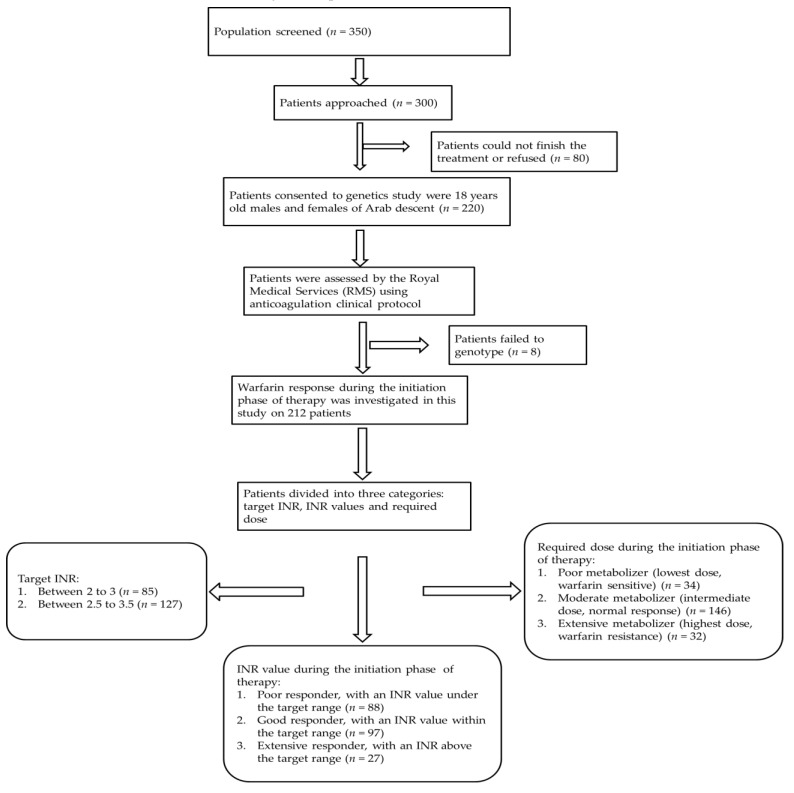
Flow chart depicting study design. INR: international normalized ratio.

**Figure 2 genes-09-00578-f002:**
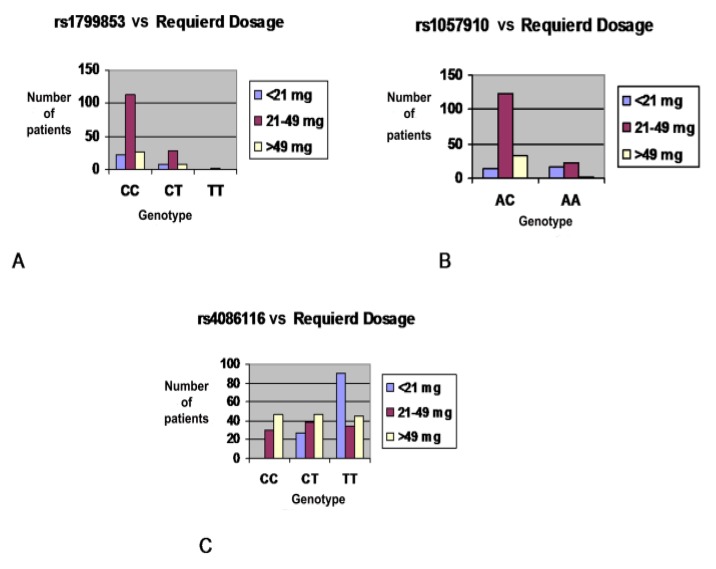
The distribution of warfarin dose by CYP2C9 genotypes during the initiation phase of therapy for 212 Jordanian cardiovascular patients: X axis represents different CYP2C9 genotypes, Y axis represents the proportion of patients across each genotype, Blue column represents a sensitive group who required the lowest warfarin dose (<21 mg/week), Purple column represents the intermediate group who required moderate warfarin dose ((21–49) mg/week), Yellow column represents a resistant group who required the highest warfarin dose (>49 mg/week). (**A**) Distribution of warfarin dose by rs1799853 variant. (**B**) Distribution of warfarin dose by rs1057910 variant. (**C**) Distribution of warfarin dose by rs4086116 variant.

**Figure 3 genes-09-00578-f003:**
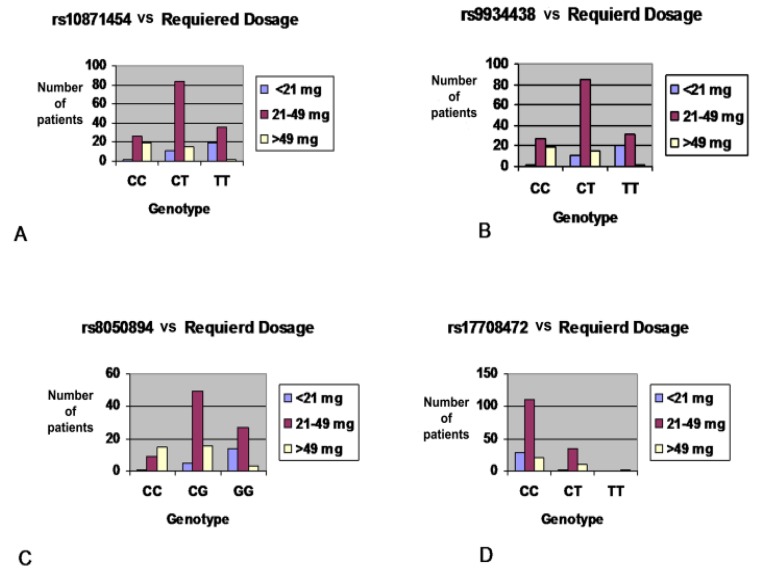
The distribution of warfarin dose by *VKORC1* genotypes during the initiation phase of therapy for 212 Jordanian cardiovascular patients: X axis represents different *VKORC1* genotypes, Y axis represents the proportion of patients across each genotype, Blue column represents a sensitive group who required the lowest warfarin dose (<21 mg/week), Purple column represents the intermediate group who required moderate warfarin dose ((21–49) mg/week), Yellow column represents a resistant group who required the highest warfarin dose (>49 mg/week). (**A**) Distribution of warfarin dose by rs10871454 variant. (**B**) Distribution of warfarin dose by rs9934438 variant. (**C**) Distribution of warfarin dose by rs8050894 variant. (**D**) Distribution of warfarin dose by rs17708472 variant.

**Table 1 genes-09-00578-t001:** Descriptive analysis of demographics and clinical characteristics of 212 cardiovascular patients treated with warfarin at the Queen Alia Heart Institute.

Category	Subcategory	Extensive Metabolizer	Good Metabolizers	Poor Metabolizers
Demographics	Patients (N, %)	(32/212) 15.1%	(146/212) 68.9%	(34/212) 16%
	Age ^a^ (years)	56.0 (17.68)	55.0 (14.64)	48.29 (15.09)
	BMI ^a^	27.87 (3.72)	27.7 (4.85)	27.42 (3.45)
	Smoking (N, %)	31.25%	18.6%	41.2%
	Male	59.4%	51.4%	67.6%
	Female	40.6%	48.6%	32.4%
Concomitant Disease	Co morbidity	56.3%	68.5%	55.9%
	Hypertension	34.4%	42.5%	23.5%
	Diabetes mellitus	18.8%	21.9%	26.5%
	CHD ^b^	28.1%	25.3%	29.4%
	Thyroid	0%	3.4%	2.9%
	Lipid	3.1%	6.8%	2.9%
Medication	Aspirin	62.5%	65.8%	76.5%
Indication of Treatment	MVR ^c^	18.8%	10.3%	20.6%
	AVR ^d^	6.3%	24.0%	20.6%
	AF ^e^	34.4%	19.2%	20.6%
	DVR ^f^	9.4%	15.8%	11.8%
	Others	9.4%	7.5%	0.0%
Target INR	2–3	43.8%	39.7%	38.2%
	2.5–3.5	56.3%	60.3%	61.8%
Mean weakly dose ^a^		16.699 (2.79)	35.896 (7.39)	67.44 (42.48)
Mean INR ^a^		2.82 (0.72)	2.38 (0.75)	2.44 (0.83)

^a^ Mean Standard deviation in square brackets. ^b^ CHD: Chronic heart disease. ^c^ MVR:Mitral valve replacement. ^d^ AVR: Aortic valve replacement. ^e^ AF: Atrial Fibrillation. ^f^ DVR: Double valve replacement.

**Table 2 genes-09-00578-t002:** Association of *VKORC1* and *CYP2C9* single nucleotide polymorphism (SNPs) with warfarin sensitivity during the initiation phase of therapy of 212 cardiovascular patients.

Gene	SNP ID	Genotype	Sensitive	Moderate	Resistance	*p*-Value *
*VKORC1*	rs10871454	CC	4.3%	57.4%	38.3%	<0.001
		CT	10.0%	76.4%	13.6%	
		TT	34.5%	63.6%	1.8%	
	rs8050894	CC	2.3%	60.5%	37.2%	<0.001
		CG	10.9%	74.5%	14.5%	
		GG	32.2%	64.4%	3.4%	
	rs9934438	CC	4.2%	58.3%	37.5%	<0.001
		CT	9.9%	76.6%	13.5%	
		TT	35.8%	62.3%	1.9%	
	rs17708472	CC	18.1%	68.8%	13.1%	<0.001
		CT	6.1%	73.5%	20.4%	
		TT	0.0%	0.0%	100%	
*CYP2C9*	rs1799853	CC	14%	70.1%	15.9%	0.744
		CT	19.6%	63%	17.4%	
		TT	0.0%	100%	0.0%	
	rs4086116	CC	8.1%	73.2%	18.7%	0.012
		CT	24.1%	62.0%	13.9%	
		TT	30.0%	70.0%	0.0%	
	rs1057910	AA	8.8%	72.5%	18.7%	<0.001
		AC	41.5%	53.7%	4.9%	

* Chi-Square Test with *p*-value < 0.05 is considered significant.

**Table 3 genes-09-00578-t003:** Frequencies of the haplotypes of *VKORC1* and *CYP2C9* genes among the 212 warfarin sensitive patients.

Gene	Haplotypes	Frequency * (%)	Odds Ratio (95% CI)	*p*-Value **
*VKORC1*	TGAG	0.512	0.00	------
	CCGG	0.324	0.32 (0.2–0.43)	<0.0001
	CCGA	0.129	0.38 (0.23–0.54)	<0.0001
	CGGG	0.028	0.34 (0.03–0.66)	0.034
	TCGG	0.007	0.21 (−0.38–0.8)	0.48
*CYP2C9*	CAC	0.767	0.00	-----
	TAT	0.116	−0.05 (−0.21–0.11)	0.53
	TCC	0.094	−0.45 (−0.64–−0.27)	<0.0001
	TAC	0.021	0.01 (−0.34–0.37)	0.95
	TCT	0.002	−1.11 (−2.14–−0.08)	0.037

* Genetic haplotype frequency of 212 warfarin intake patients, ** *p*-value < 0.05 is considered significant.

**Table 4 genes-09-00578-t004:** Association of *VKORC1* and *CYP2C9* SNPs with variability on warfarin required doses and with INR treatment outcome.

SNP ID	Initiation Dose	*p*-Value *	Initiation INR	*p*-Value *
rs10871454	38.1 (23.02)	<0.001	2.46 (0.77)	0.006
rs8050894		<0.001		0.008
rs9934438		<0.001		0.009
rs17708472		<0.001		0.511
rs1799853		0.118		0.184
rs4086116		0.001		0.08
rs1057910		0.001		0.572

* Kurskal Wallis test with *p*-value < 0.05 is considered significant, Mean Standard deviation in square brackets.

**Table 5 genes-09-00578-t005:** Association of *VKORC1* and *CYP2C9* SNPs with response to warfarin during the initiation phase of therapy of 212 cardiovascular patients.

Gene	SNP ID	Genotype	Poor Responder	Good Responder	Extensive Responder	*p*-Value *
*VKORC1*	rs10871454	CC	55.3%	36.2%	8.5%	0.171
		CT	40.9%	45.5%	13.6%	
		TT	30.9%	54.5%	14.5%	
	rs8050894	CC	53.5%	39.5%	7%	0.235
		CG	41.8%	43.6%	14.5%	
		GG	32.2%	54.2%	13.6%	
	rs9934438	CC	54.2%	37.5%	8.3%	0.226
		CT	40.5%	45%	14.4%	
		TT	32.1%	54.7%	13.2%	
	rs17708472	CC	38.1%	50.6%	11.3%	0.042
		CT	55.1%	28.6%	16.3%	
		TT	0.0%	66.7%	33.3%	
*CYP2C9*	rs1799853	CC	45.1%	44.5%	10.4%	0.076
		CT	28.3%	52.5%	19.6%	
		TT	50.0%	0.0%	50.0%	
	rs4086116	CC	45.5%	43.9%	10.6%	0.005
		CT	39.2%	49.4%	11.4%	
		TT	10.0%	40.0%	50.0%	
	rs1057910	AA	42.1%	45.0%	12.9%	0.910
		AC	39%	48.8%	12.2%	

* Chi-Square Test with *p*-value < 0.05 is considered significant.

**Table 6 genes-09-00578-t006:** Frequencies of the haplotypes of *VKORC1* and *CYP2C9* genes among the 212 warfarin responsiveness patients.

Gene	Haplotypes	Frequency * (%)	Odds Ratio (95% CI)	*p*-Value **
*VKORC1*	TGAG	0.512	0.00	------
	CCGG	0.326	−0.18 (−0.33–−0.03)	0.02
	CCGA	0.129	−0.08 (−0.28–0.12)	0.46
	CGGG	0.026	−0.05 (−0.47–0.36)	0.8
	TCGG	0.007	0.55 (−0.22–1.32)	0.16
*CYP2C9*	CAC	0.767	0.00	------
	TAT	0.115	0.25 (0.04–0.45)	0.018
	TCC	0.094	0.06 (−0.17–0.3)	0.59
	TAC	0.021	0.44 (−0.01–0.89)	0.059
	TCT	0.003	0.4 (−0.9–1.71)	0.55

* Genetic haplotype frequency of 212 warfarin intake patients, ** *p*-value < 0.05 is considered significant.

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
