# Peer review of "Impact of CYP2C9 and VKORC1 Polymorphisms on Warfarin Sensitivity and Responsiveness in Jordanian Cardiovascular Patients during the Initiation Therapy"

_genes, 2018, doi:10.3390/genes9120578_

Round 1

Reviewer 1 Report

Dear authors!

Thank you for presenting results of your study. I think that the article must be improved significantly. 

1. Introduction. There were plenty of systematic reviews and meta-analyses about warfarin's pharmacogenetics and inter-ethnic variability. Examples: 

PMID: 27073641; PMID: 28025970; PMID: 15714076; PMID: 19942260... But there was no any reference in your article... Moreover, you cited extremely old original studies (1994 and so on) - is it so important? I recommend you to base on fresh publications.

2. Your approach was observational study. Unfortunately, it has low level of evidence, and there were conducted a lot of studies of warfarin's pharmacogenetics with similar design 5-10-15 years ago. Yes, it is novel - to study Jordanian population, there was no analogous studies early. But you need to describe how Jordanian descent was determined - was it self-identification of participant and his parents? Or just of participant? Or it was not self-identification?

3. Check your sample for Shapiro-Wilk test of distribution's normality... You made ANOVA, but did not presented basis for that test's acceptability.

4. Discussion. Yes, there was associations of polymorphisms with outcomes. So, please do not write "increase", suggest - "was associated with incresed..." and so on.

5. You have several SNPs for one gene... I strongly recommend you to make haplotype analysis (for example, you can use "SNPStat" free software). Without that your study seems to be unfinished.

6. Material and methods. You divided patients according to warfarin responding by INR... But there was no adjustment by dose and by time to reach target INR! It is important to describe, particularly if you have medical documentation. And results in Table 4 must be adjusted. 

Author Response

Dear Editor in Chief,

            I would like to extend my deepest thanks to the reviewers for their constructive comments and suggestions with regard to the manuscript titled “Impact of CYP2C9 and VKORC1 polymorphisms on warfarin sensitivity and responsiveness in Jordanian cardiovascular patients during the initiation therapy”. I am pleased to submit the revised version of the paper that addresses each one of the reviewer comments. The manuscript was also reviewed by an English language editor in order to enhance its flow and the communicability of its scientific content.

Comments by Reviewer #1

1-      Introduction. There were plenty of systematic reviews and meta-analyses about warfarin's pharmacogenetics and inter-ethnic variability. Examples: 

PMID: 27073641; PMID: 28025970; PMID: 15714076; PMID: 19942260... But there was no any reference in your article... Moreover, you cited extremely old original studies (1994 and so on) - is it so important? I recommend you to base on fresh publications.

The comment considered and the introduction updated according to the systematic   reviews.

2-      Your approach was observational study. Unfortunately, it has low level of evidence, and there were conducted a lot of studies of warfarin's pharmacogenetics with similar design 5-10-15 years ago. Yes, it is novel - to study Jordanian population, there was no analogous studies early. But you need to describe how Jordanian descent was determined - was it self-identification of participant and his parents? Or just of participant? Or it was not self-identification?

It was self-identification of participant and his parents and each participant was asked to complete a form containing his\her name up to five generations to ensure the purity of their lineage.  

3-      Check your sample for Shapiro-Wilk test of distribution's normality... You made ANOVA, but did not presented basis for that test's acceptability

According to Shapiro-Wilk test our data were not normaly distrbuted (datat not shown), therfore, we conducted Kurskal Wallis test (non parmetric test used with non normaly distrbuted data) and therfore, p-values within Table 4 was rewritten.

4-      Discussion. Yes, there was associations of polymorphisms with outcomes. So, please do not write "increase", suggest - "was associated with incresed..." and so on.

      Suggestion was applied  

5-      You have several SNPs for one gene... I strongly recommend you to make haplotype analysis (for example, you can use "SNPStat" free software). Without that your study seems to be unfinished

           Suggestion was applied and the haplotype analysis can be found in Table 3 and 6.

            6-      Material and methods. You divided patients according to warfarin responding by                               INR... But there was no adjustment by dose and by time to reach target INR! It is                               important to describe, particularly if you have medical documentation. And results in                        Table 4 must be adjusted

                    We divided the patients into two groups according to the INR values (responder                                group, Table 5) and according to the initiation dose (sensitive group, Table 2), but we                        did not consider the time to reach the target INR, but we will take it in consideration in                      the future research. (Refer to section 2.2 of the Material and Methods), Outcome                              Measure (lines 102-105, responder group according to the mean of the initiation INR                        values and lines 106-111 for the sensitive group according to the mean of warfarin                            initiation dose)).

Comments by Reviewer # 2

1-      In “Materials and Methods “should be mentioned if the isoenzyme CYP2C9 strong inhibitors or inductors was excluded in pharmacotherapy during the study.

Yes, it was excluded during this study, this information was added to the exclusion criteria within materials and methods section.

             2-      In “Discussion” should be mentioned, that also less important polymorphisms are                            involved in warfarin efficacy and metabolism, i.e. OATP transporters (facilitating                                warfarin transport to hepatocyte), isoenzymes CYP1A1 and CYP1A2 (R-warfarin                             biodegradation), or vitamin K–dependent carboxylase polymorphism. These factors                          was not studied in the present work.

            These factors will be considered as a future direction in further research.

Please do not hesitate to contact me if you need any additional information. I highly look forward to hearing from you.

Yours sincerely,

Dr. Laith N. AL-Eitan, PhD
Associate Professor of Human Genetics and Pharmacogenetics

Department of Biotechnology & Genetic Engineering  
Faculty of Science and Arts 
Jordan University of Science and Technology,
P.O.Box 3030, Irbid 22110, JORDAN
Email: [email protected] 
Cellphone: +962772322011

Reviewer 2 Report

In “Materials and Methods“ should be mentioned if the isoenzyme CYP2C9 strong inhibitors or inductors was excluded in pharmacotherapy during the study.

In “Discussion” should be mentioned, that also less important polymorphisms are involved in warfarin efficacy and metabolism, i.e. OATP transporters (facilitating warfarin transport to hepatocyte), isoenzymes CYP1A1 and CYP1A2 (R-warfarin biodegradation), or vitamin K–dependent carboxylase polymorphism. These factors was not studied in the present work.

Author Response

(The authors gave the same response as above.)

Round 2

Reviewer 1 Report

Thank you! I recommend to accept the article